# Gut Microbiome in Post-COVID-19 Patients Is Linked to Immune and Cardiovascular Health Status but Not COVID-19 Severity

**DOI:** 10.3390/microorganisms11041036

**Published:** 2023-04-15

**Authors:** Olga N. Tkacheva, Natalia S. Klimenko, Daria A. Kashtanova, Alexander V. Tyakht, Lilit V. Maytesyan, Anna A. Akopyan, Stanislav I. Koshechkin, Irina D. Strazhesko

**Affiliations:** 1The “Russian Clinical Research Center for Gerontology” of the Ministry of Healthcare of the Russian Federation, Pirogov Russian National Research Medical University, 16 1st Leonova Str., 129226 Moscow, Russia; 2Atlas Biomed Group—Knomx LLC, Tintagel House, 92 Albert Embankment, Lambeth, London SE1 7TY, UK; 3Center for Precision Genome Editing and Genetic Technologies for Biomedicine, Institute of Gene Biology Russian Academy of Sciences, 34/5 Vavilova Str., 119334 Moscow, Russia

**Keywords:** COVID-19, post-COVID syndrome, gut microbiome, nearest balance, SARS-CoV-2

## Abstract

The composition of the gut microbiome stores the imprints of prior infections and other impacts. COVID-19 can cause changes in inflammatory status that persist for a considerable time after infection ends. As the gut microbiome is closely associated with immunity and inflammation, the infection severity might be linked to its community structure dynamics. Using 16S rRNA sequencing of stool samples, we investigated the microbiome three months after the end of the disease/infection or SARS-CoV-2 contact in 178 post-COVID-19 patients and those who contacted SARS-CoV-2 but were not infected. The cohort included 3 groups: asymptomatic subjects (*n =* 48), subjects who contacted COVID-19 patients with no further infection (*n =* 46), and severe patients (*n =* 86). Using a novel compositional statistical algorithm (nearest balance) and the concept of bacterial co-occurrence clusters (coops), we compared microbiome compositions between the groups as well as with multiple categories of clinical parameters including: immunity, cardiovascular parameters and markers of endothelial dysfunction, and blood metabolites. Although a number of clinical indicators varied drastically across the three groups, no differences in microbiome features were identified between them at this follow-up point. However, there were multiple associations between the microbiome features and clinical data. Among the immunity parameters, the relative lymphocyte number was linked to a balance including 14 genera. Cardiovascular parameters were associated with up to four bacterial cooperatives. Intercellular adhesion molecule 1 was linked to a balance including ten genera and one cooperative. Among the blood biochemistry parameters, calcium was the only parameter associated with the microbiome via a balance of 16 genera. Our results suggest comparable recovery of the gut community structure in the post-COVID-19 period, independently of severity or infection status. The multiple identified associations of clinical analysis data with the microbiome provide hypotheses about the participation of specific taxa in regulating immunity and homeostasis of cardiovascular and other body systems in health, as well as their disruption in SARS-CoV-2 infections and other diseases.

## 1. Introduction

The coronavirus infection (COVID-19) is spreading around the world. More than 6.82 million people worldwide have died from COVID-19 since its first detection in December 2019 (according to the World Health Organization, COVID-19 Weekly Epidemiological Update Edition 133 published 8 March 2023). Against the background of a pandemic and a large difference in the severity of the disease after infection, an important issue today is the study of consequences of the infection.

While SARS-CoV-2 primarily causes lung infection by binding to ACE2 receptors [1] present on alveolar epithelial cells, it has recently been reported that SARS-CoV-2 RNA had been found in the feces of infected patients [2]. Notably, intestinal epithelial cells, in particular enterocytes of the small intestine, also express ACE2 receptors [3,4]. Some patients with COVID-19 have experienced diarrhea, which indicates a clear possibility of the involvement of the gut-lung axis in the disease process [5]. Moreover, observational evidence suggests that SARS-CoV-2 can infect and be shed from the gastrointestinal tract [6].

The role of the gut microbiota in influencing various lung diseases is being studied not only in the context of coronavirus infections. Results have already been obtained showing a relationship between the microbiota composition and the influenza virus, respiratory syncytial virus [7], and acute respiratory distress syndrome [8]. Respiratory viral infections themselves are also known to cause changes in the gut microbiota [9]. In addition to these associations, the gut microbiome plays a critical role in the functioning of major components of the host’s innate and adaptive immune systems [10]. Pronounced shifts in the gut microbiota composition are associated with a wide range of pathologies, including autoimmune, allergic, and chronic inflammatory diseases [11,12,13], such as inflammatory bowel disease [14], colorectal and other cancer types [15], type 2 diabetes [16], cardiovascular disease [17], neurodegenerative diseases [18], and even depression [19]. The depletion of the gut microbiota with age, namely a decrease in its diversity [8,20], is a well-known phenomenon which can potentially can affect the severity of COVID-19 through changes in the functioning of the immune system, as happens with other infections. For example, in mouse models, the removal of certain intestinal bacteria with antibiotics leads to increased susceptibility of the lungs to influenza virus infection [21]. However, according to the recent data, antibiotics could have an opposite or even antiviral effect [22].

Recent study of blood biomarkers from COVID-19 patients have identified a set of molecular predictors (blood proteomic biomarkers) that may characterize and predict individual differences in disease severity [23]. One study found that a core of gut microbiota can accurately predict the aforementioned proteomic biomarkers, and these features of the gut microbiota are highly correlated with pro-inflammatory cytokines [24].

A recent small study in COVID-19 patients [25] showed that gut microbiome changes during hospitalization were associated with fecal levels of SARS-CoV-2 and severity of COVID-19. Among the most pronounced changes were enrichment in opportunistic microorganisms and depletion of beneficial bacteria. Moreover, this imbalance persisted even after the elimination of SARS-CoV-2 and the disappearance of respiratory symptoms. This suggests that SARS-CoV-2 infection could be associated with long-term adverse effects on the gut microbiota [25]. Another recent study showed that fecal microbial composition differed significantly between SARS-CoV-2 patients and controls, independently of antibiotic exposure [26]. Some opportunistic bacteria were enriched in COVID-19 patients compared to controls. However, there were no differences in microbial community structure between recovered patients and non-infected controls, nor a difference in alpha diversity between all groups. No significant associations were found between microbiome composition and disease severity [26]. Chen et al. showed that microbiota diversity had not returned to initial levels six months after recovery, and patients with lower gut microbiome diversity showed higher inflammation level and illness severity during the acute phase [27]. Interestingly, post-COVID syndrome was associated with an initial gut microbiome rich in opportunistic pathogens [28].

It is important to explore whether there are persistent alterations in gut microbiome after recovery and if the microbiome can contribute to the severity of post-COVID-19 symptoms. For this purpose, we conducted an observational study of three subgroups of patients characterized by strict exclusion/inclusion criteria, clearly defined clinical status, and a detailed panel of diverse laboratory tests.

## 2. Methods

### 2.1. Trial Information

The study was conducted from January 2021 to May 2021 at the Russian Gerontological Research and Clinical Center (ClinicalTrials.gov Identifier: NCT04871789), and was approved by the local ethics committee under protocol No. 36 dated 2 November 2020.

Originally, a total of 200 subjects were enrolled into the study, consisting of 50 individuals who had recovered from asymptomatic COVID-19, 50 individuals who had been in close contact for at least three days with patient(s) with confirmed COVID-19 infections but did not contract the virus and had no IgM (N-protein and RBD S-protein) and IgG (whole S-protein) antibodies to SARS-CoV-2 in their serum (Vektor-Best, Novosibirsk, Russia), and 100 individuals who had recovered from severe COVID-19. Blood analyses were performed for all 200 subjects, while stool samples were collected from a slightly lower number of subjects, including 48 asymptomatic subjects (Group A), 46 subjects with no further infection (Group N), and 86 severe patients (Group S). These were the groups that were analyzed in the present study.

### 2.2. General Recruitment Criteria

All participants were at least 18 years of age and provided signed informed consent forms. The inclusion criteria were group specific, as follows.

Group A (asymptomatic):Confirmed COVID-19 diagnosis based on:○A medical record of a positive RT-PCR test result for SARS-CoV-2; or○IgM and IgG antibody titers against SARS-CoV-2 in the serum, indicating that the participant has been previously infected with SARS-CoV-2.No history of clinical symptoms associated with COVID-19 in the past six months, including fever (body temperature above 37.5 degrees Celsius/99.5 degrees Fahrenheit), shortness of breath, smell and taste dysfunctions, diarrhea, and coughing.

Group N (non-infected):Previous close contact for at least three days without personal protective equipment with individuals who exhibited COVID-19 symptoms (at home, in the workplace, etc.)No IgM and IgG antibody titers against SARS-CoV-2 in the serum, indicating that the participant has not been previously infected with SARS-CoV-2No history of clinical symptoms associated with COVID-19 for the past six months, including fever (body temperature above 37.5 degrees Celsius/99.5 degrees Fahrenheit), shortness of breath, smell and taste dysfunctions, diarrhea, and coughing.

Group S (severe):Confirmed COVID-19 diagnosis based on:○A medical record of a positive RT-PCR test result for SARS-CoV-2; or○IgM and IgG antibody titers against SARS-CoV-2 in the serum, indicating that the participant has been previously infected with SARS-CoV-2.Inpatient treatment of severe COVID-19

Patients with COVID-19 were considered to have severe illness if they exhibit one or more of the following symptoms:Respiratory rate > 30 breaths/minBlood oxygen saturation (SpO2) ≤ 93%PaO2/FiO2 < 300Computed tomography (CT) findings of lung damage > 50%Septic shockMultiple organ failureCytokine storm

The following exclusion criteria were identical across the groups:PregnancyRefusal to participatePrior COVID-19 vaccination

The following procedures were carried out over the course of two visits.

Visit 1 (screening), days 1–14:Signing of an informed consent formExpress pregnancy test (for women of childbearing age)Measurement of IgM and IgG antibody titers against SARS-CoV-2Intake of medical and epidemiological histories

Visit 2, day 1 for 90 ± 15 patients since diagnosis or discharge from the hospital (for severe COVID-19 patients):

All groups:An extensive survey and intake of detailed life and medical historiesPhysical examination, including measurement of arterial blood pressure (ABP), heart rate (HR), respiratory rate (RR), and anthropometric status indicators including hip and waist circumference and body mass index (BMI)Complete blood count with white blood cell differentialBlood chemistry test, including homeostasis, immune status, level of sex hormones, and hormonal markers of metabolic dysfunctionsBioelectrical impedance analysisStool sampling for gut microbiota analysis by 16S rRNA gene sequencing (Group A, 45 samples; Group N, 50 samples; Group S, 88 samples)Express smell test

Group S:Chest CTSpirometrySpeckle tracking echocardiography

### 2.3. Sample Collection and Processing

Stool samples were collected using DNA-stabilizing KnomX gut microbiome collection kits (KnomX, Moscow, Russia) stored at −30 °C. DNA extraction from the stool samples was carried out using the Qiagen Power Fecal PRO kit according to the manufacturer’s instructions. Amplification of the V4 region of the 16S rRNA gene was performed using modified 515F and 805R primers. The second round of amplification was performed using standard Illumina indexes with adapters. Both rounds of PCR were performed using the Eurogen PCR buffer and the Bio-Rad CFX-96 amplifier. PCR products were purified using the Cleanup Mini kit for DNA extraction (Evrogen, Moscow, Russia). The DNA concentration was determined using a Qubit fluorometer (Invitrogen, Waltham, MA, USA) and the Quant-iT dsDNA High-Sensitivity Assay Kit. Purified amplicons were mixed equimolarly according to the obtained concentrations. Further preparation of the samples for sequencing and sequencing of the pooled library was performed using the MiSeq Reagent Kit v2 (500 cycles) and the MiSeq sequencer (Illumina, San-Diego, CA, USA) according to the manufacturer’s recommendations. Primary processing (barcode extraction) was performed as previously described [29]. After quality trimming, reads were merged using the SeqPrep package; the total length of the merged reads was 252 bp.

### 2.4. Data Analysis

The data were analyzed using the Knomics-Biota analytical system (https://biota.knomx.com/, accessed on 1 March 2023) [30]. The reads were quality-trimmed and filtered using the QIIME v2 software package [31]. Denoising of reads was conducted using the DADA2 algorithm [32]. Taxonomic classification of the obtained amplicon sequence variants (ASV) was performed using a classifier implemented in the QIIME2 software package [33] preliminary trained with the SILVA v.138 database [34], which was preprocessed using RESCRIPt (https://github.com/bokulich-lab/RESCRIPt, accessed on 8 March 2023), Creative Commons Attribution 4.0 License (CC-BY 4.0)). The 16S rRNA sequences in the database were trimmed according to the primers used (515F/806R) [35] and aggregated with a similarity threshold of 99%. Microbial abundance tables at the levels of species, genus, family, etc., were obtained by summing the relative abundance levels of ASV belonging to the corresponding clade.

Alpha diversity was evaluated using Shannon [36] and Chao1 [37] metrics for ASV abundance tables rarified to 10,459 reads per sample (the lowest number of reads per sample across all samples). Beta diversity was evaluated using rarefied genera abundance levels with the Aitchison distance metric [38]. During clr-transformation (centered log-ratio transformation) of microbial abundances, zero taxa counts were replaced with a pseudocount (0.5). Co-abundance networks of microbial genera were obtained using SPIEC-EASI algorithm [39] with the Meinshausen-Bühlmann method for correlations detection (other parameters: number of subsamples—10, number of lambda iterations—10, minimum value of lambda—0.2). Only the genera with abundance > 20 reads in > 10 samples were considered in this analysis [40,41]. Clusters of co-abundant genera (microbial cooperative; shortly—coops) were derived from the resulting network using the Louvain method [42].

We used a similar statistical approach to compare the gut microbiome composition across three groups of subjects, as well as with multiple clinical parameters. To reduce the analysis dimensionality, we selected one parameter from each cluster of highly correlated parameters (Spearman correlation coefficient > 0.8, see Appendix A). There were four groups of parameters:Laboratory tests related to immune status (*n* = 18)Clinical cardiovascular markers (*n* = 21)Laboratory markers of endothelial dysfunction (*n* = 5)Blood metabolites (*n* = 21)

The comparison of microbiome composition with the study groups and clinical parameters included: analysis of alpha diversity using a linear model, beta diversity using PERMANOVA for categorical factors, and distance-based redundancy analysis (dbRDA) for continuous factors (using the adonis function from the package vegan [43]), as well as analysis of microbial cooperatives using a compositionality-aware approach (see below). For the clinical parameters, correction for multiple comparisons was performed using the false discovery rate (FDR) calculation with the Benjamini-Hochberg method. The correction was performed separately for each of the five parameter groups described above. For each parameter significantly associated with beta diversity, a detailed analysis of the association was performed using the nearest balance method [44] as follows. Balance is a normalized ratio of ≥1 microbial taxa in the numerator to ≥1 taxa in the denominator used to represent microbial abundance data in a compositional manner. Half of the samples were randomly selected from the data 100 times. Each time, the nearest balance associated with the analyzed factor was identified as described in [44]. The genera assigned to numerator in >90 iterations or to denominator in >90 iterations were included in the final balance.

Associations of microbiome composition represented as cooperatives of microbial genera were also explored from the perspective of balances. For each coop, we composed a balance including its genera in the numerator and all other genera in the denominator. The associations of these balances with the clinical parameters were evaluated using a linear model. A correction for multiple comparisons was performed using the Benjamini-Hochberg algorithm.

## 3. Results

### 3.1. Gut Microbiome Composition Is Not Associated with the Severity of Disease after Three Months

The overall microbiome composition of the analyzed cohorts was similar to those previously described for stool profiles of a geographically similar population [45]. The top three dominant genera were *Bacteroides* (mean relative abundance: 22%), *Prevotella* (10%), and *Faecalibacterium* (9%).

The community structure was not different three months after infection in asymptomatic COVID-19 group, after contact with the SARS-CoV-2 virus in the resistant group, or three months after the end of the disease for the severe group. Specifically, the microbiome composition was not different between subjects who had contact with COVID-19 patients but did not get infected (“resistant”) (N, *n* = 46), had asymptomatic COVID-19 (A, *n* = 48), and had severe COVID-19 (S, *n* = 86). The microbiome composition was not associated with IgM antibodies levels to the SARS-CoV-2 N protein or IgG antibodies to the S1 and S2 proteins. This analysis included testing for differences in alpha diversity (linear model, for groups—*p* > 0.5, for antibodies—*p* > 0.4), beta diversity (for groups—PERMANOVA *p* > 0.4, for antibodies—dbRDA *p* > 0.7; Figure 1), and differential abundance analysis for balances of genera cooperatives (linear model, for groups—FDR > 0.4, for antibodies—FDR > 0.8), as well as clr-transformed abundance of taxa aggregated to ranks from species to phylum (FDR > 0.4 and 0.9, respectively).

While no microbiome differences were found between the groups, the analysed cohort had been deeply phenotyped three months after the infection, and many non-microbiome-related clinical factors were found to be significantly different between the three groups. For many of these factors, their values were notably outside the reference ranges. This excessive variability provided an opportunity to investigate the links between microbiome composition and a greater statistical power. Specifically, three groups of clinical factors were analyzed: immune status, cardiovascular system parameters, and blood biochemistry parameters.

### 3.2. Balance of Specific Bacterial Taxa Is Linked to Immunity via Relative Level of Lymphocytes

Among the immune status parameters (*n* = 18), only the relative number of the lymphocytes was significantly associated with general microbiome composition (dbRDA, FDR = 0.0643, R^2^ = 0.0116). The parameter values exceeded the reference range for approximately one third of the patients (Figure 2B). The revealed association was investigated in detail using the nearest balance method [44] at the genus level. In one of its scenarios, this novel compositional-aware method allows identifying the balance (normalized ratio of two groups of microbial taxa—numerator and denominator) that is optimally associated with an external factor of choice. During the cross-validation procedure, we identified seven reproducible genera (reproducibility > 90%) in the nearest balance numerator and seven in the denominator. Among them, the highest positive link was observed for the unclassified members of the RF39 order, Clostridia UCG014, Oscillospirales UCG010, as well as the *Akkermansia* genus from the numerator. The strongest negative associations were found in the *Parasutterella*, *Flavonifractor* genera and the [*Ruminococcus*] *gnavus* group located in the denominator (Figure 2A). Neither alpha diversity nor genera cooperatives were significantly associated with the parameters from this group.

### 3.3. Links between Microbiome and Clinical Cardiovascular and Endothelial Dysfunction Markers

Because COVID-19 is known to have a significant impact on the cardiovascular system, we compared the gut microbiome profiles with the parameters from the following groups of clinical cardiovascular markers (*n* = 21, Table 1): pulse pressure and heart rate measured in sitting position, as well as the results of carotid Doppler ultrasonography (CDU, 2 parameters), cardiac ultrasound (CU, 13 parameters) and applanation tonometry (AT, 4 parameters).

No significant associations were observed in alpha or beta diversity. Out of the 44 explored clinical cardiovascular markers, six were significantly associated with genera cooperatives (Figure 3, Table 2, FDR < 0.05). At the same time, no significant associations between alpha diversity or microbiome composition in general were revealed (dbRDA) for these parameters (FDR > 0.1).

Interestingly, all factors for which there were significant associations with detected cooperatives were positively correlated between each other (Spearman rho—from 0.17 to 0.54, FDR < 0.04, Figure 3).

Separately, we investigated if the microbiome was related to the laboratory parameters of the markers of endothelial dysfunction (*n* = 5; group statistics are listed in the Table 3), including the following:-Vascular endothelial soluble growth factor receptor 1 (VEGF-R1)-Intercellular adhesion molecule 1 (ICAM-1)-Vascular endothelial adhesion molecule type 1 (VCAM-1)-E-selectin-Von Willebrand factor

Among them, ICAM-1 was significantly associated with the general microbiome composition (dbRDA, FDR = 0.0314, R^2^ = 0.0374) and was negatively associated with the *Faecalibacterium* cooperative, including *Faecalibacterium* itself and the *Lachnospiraceae* NK4A136 group. The association with the general microbiome composition was further investigated by nearest balance cross-validation analysis (see Section 2). The analysis revealed 10 reproducible taxa, among which the strongest positive associations were observed for *Romboustia* and [*Ruminococcus*] *gnavus*, and strongest negative ones were observed with *Lachnospiraceae* groups UCG-010 and NK4A136, *Barnesiella*, and *Eubacterium xylanophillum* (Figure 4).

### 3.4. Calcium Is the Blood Metabolite Most Strongly Associated with Gut Microbiome

Among the 21 analyzed blood metabolites, there were no significant associations found within the microbiome. The highest significance (lowest p value) was observed for the total blood calcium level in relation to the general microbiome composition (dbRDA, FDR = 0.0914, R^2^ = 0.0116). We further explored this trend in a targeted manner using the nearest balance approach. Nearest balance cross-validation revealed 16 reproducible genera associated with the total calcium level (Figure 5). The topmost positively associated genera included *Erysipelatoclostridium* and [*Ruminococcus*] *gnavus*, and the topmost negatively associated genera were [*Eubacterium*] *siraeum* and *Methanobrevibacter* archaeon.

## 4. Discussion

Despite the initial hypothesis, three months after encountering the SARS-CoV-2 virus, no microbiome signature was found to distinguish between resistant and non-resistant subjects or to be associated with the severity of the disease (at least for the sample of the provided size). Although the gut microbiota appeared to be significantly disrupted during the acute COVID-19 period and in the short term after the infection [46], it is likely that after this period the gut community will essentially recover. A recent small study showed some differences in the microbiome composition of healthcare workers three months after COVID-19 recovery and a control group [47], but none of the results were replicated in our sample. A study by Zhang, F. et al. revealed a lowered functional potential to produce short-chain fatty acids in patients with severe COVID-19 30 days after recovery [48]. Nevertheless, we observed considerable long-term clinical consequences of coronavirus infections, the severity of which correlated with the composition of the intestinal microbiota.

In our study, we used a novel composition-aware method for microbiome analysis—the nearest balance [44]. Although statistical methods that consider compositionality are relatively novel in the field of microbiome and less widespread than component-based ones (i.e., those operating with percentages and relative abundance), they are more precise and, provide interpretable results in the form of microbial balances. Each balance includes one or more taxa in the numerator and one or more taxa in the denominator, highlighting two microbial groups that are positively and negatively linked to a certain factor, respectively. We discovered interesting associations of balances for multiple groups of clinical factors.

In terms of the immune status parameters, only one parameter was found to be linked to the microbiome—the relative lymphocyte level. It is well-known that the ratio of lymphocytes to neutrophils is a key markers of prognosis in patients with viral infections and other pathologies [49]. A higher relative abundance of lymphocytes could be a favorable prognostic factor, reflecting a persistent immune balance. In this study, the numerator of the balance linked to this parameter included the taxa linked to beneficial effects on human metabolism: *Akkermansia* [50] and *Christensenellaceae* [51]. The denominator of the balance included [*Ruminococcus*] *gnavus*, which has been previously related to inflammatory bowel disease [52], as well as cardiovascular risks [53]. *Parasutterella*, which was included nearby, is also considered to be associated with chronic intestinal inflammation [54]. Another interesting taxon in the denominator was *Flavonifractor*—known to be able to metabolize catechins and suppress Th2 lymphocytes’ immune response [55].

Currently, there are many diverse observations indicating the role of microbiota in the regulation of cardiovascular parameters. These observations include a reduced risk of CVD associated with higher fiber intake in prospective studies, altered gut microbiome in animal models of hypertension, as well as alterations in cardiovascular markers in germ-free animals and animals after microbiome transplantation from the hypertension model ones [56]. Furthermore, cross-sectional studies link the content of the gut microbiome to cardiovascular markers in humans [57,58,59]. Possible mechanisms considered to underlie these associations include the production of specific metabolites by the gut microbiome (short-chain fatty acids, TMAO, H_2_S), endotoxemia, and microbiome-mediated links to immunity [56,60].

Interestingly, one of the reproducible associations is a positive link between *Prevotella* abundance and impaired cardiovascular parameters (hypertension or high CVD risk) [46,48]. In our study, we observed a positive association between *Prevotellaceae*-dominated microbial cooperative and pulse pressure, as well as with the pulse wave velocity measured via applanation tonometry. Our previous study showed a correlation between *Bacteroides* and pulse wave velocity in healthy individuals [61], each previously associated with low-grade inflammation and the development of chronic diseases [62,63,64]. Arterial wall stiffness is one of the basic components of aging and cardiovascular pathologies, from increased blood pressure to chronic heart failure.

Notably, the butyrate-producing *Subdoligranulum* genus was less abundant in the subjects with higher pulmonary artery pressure and greater arterial stenosis in our study. These observations align very well with previous findings [65] regarding the protective role of these bacteria against obesity and obesity-related features. The identified negative association of *Methanobrevibacter* with end-diastolic volume is of certain interest. The data concerning these archaea and their association with human health evidence are quite contradictory [66,67,68]. A recent study showed their depletion in patients with high triglycerides levels [69], while other studies revealed their strong association with cardiovascular risk factors [70] and obesity [66].

Furthermore, our results showed that microbiota composition was associated with ICAM-1, the expression of which contributes to the clinical manifestations of a variety of diseases [71], predominantly by interfering with normal immune function in oncology, cardiovascular, and autoimmune diseases. Not surprisingly, the level of the molecule was inversely associated with the butyrate-producing and anti-inflammatory *Faecalibacterium* cooperative.

One of the puzzling discoveries was the link between blood calcium level and a microbial balance including 16 genera. The numerator of the balance included two genera reportedly associated with disorders: [*Ruminococcus*] *gnavus* group (discussed above) and *Erysipelatoclostridium*. A representative species of the latter is an opportunist microorganism, *E. ramosum* (previously *Clostridium ramosum)* that has been reported as a frequent cause of bacteremia and a contributor to high-fat diet-induced obesity [72]. It would be interesting to confirm this suggested positive association of calcium levels with the opportunistic microorganisms on a larger cohort, especially considering that not only hypocalcemia, but also hypercalcemia was associated with a poor COVID-19 prognosis [73].

The results of our study suggest that the gut microbiota had the potential to recover after SARS-CoV-2 infection, and ultimately, there were no significant differences in microbiota composition in patients with varying degrees of COVID-19 severity in the long-term. However, the identified specific novel associations between microbiota features and health state contribute to understanding the complex interactions within the holobiont system, and will allow for the development of novel strategies for microbiome-tailored disease prevention and treatment.

## Figures and Tables

**Figure 1 microorganisms-11-01036-f001:**
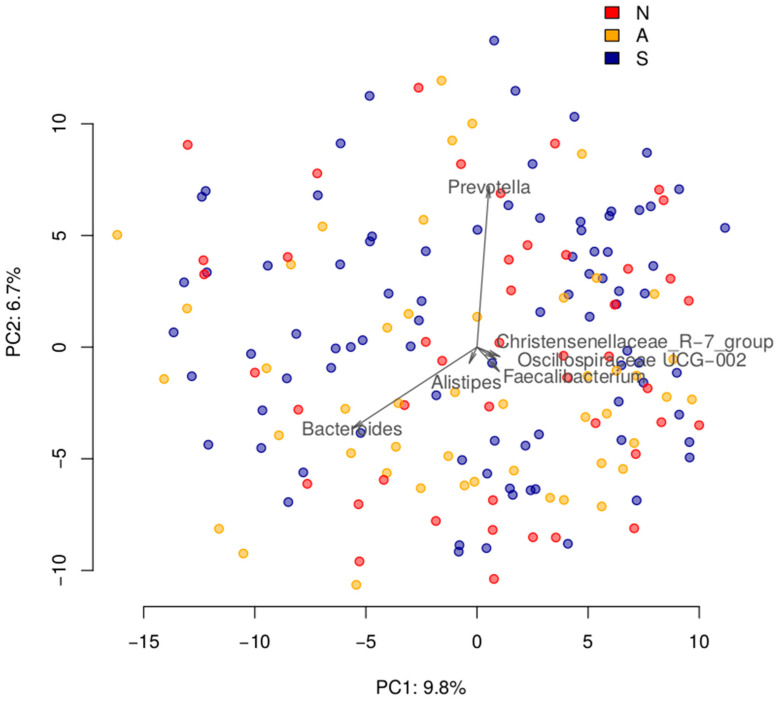
Overall comparison of post-COVID gut microbiomes across the three groups of patients with different courses of infection. The samples are visualized using Principal Coordinates Analysis (PCoA) based on the Aitchison distance. Arrows show the top taxa in terms of the explained variance in given axes, with the length proportional to the variance explained by the taxon. The arrows’ angle reflects the distribution of variance between the axes. N—non-infected group, A—asymptomatic group, S—severe group.

**Figure 2 microorganisms-11-01036-f002:**
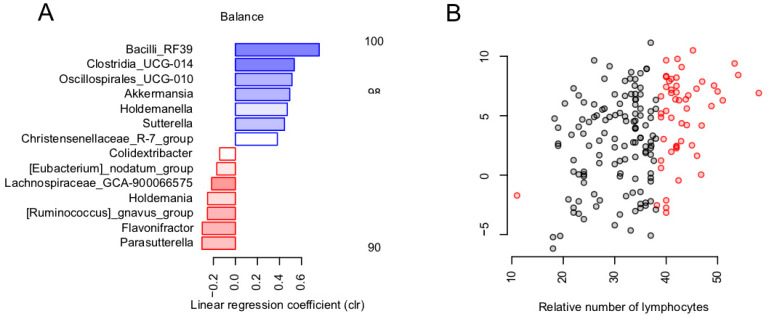
Links between immunity and gut microbiome post-COVID-19: reproducible taxa revealed by the nearest balance cross-validation associated with lymphocytes relative levels (%). (**A**) Bacterial genera—reproducible members of the nearest balances revealed in cross-validation. The upper green bars show the taxa in the numerator in decreasing order of the linear regression coefficient between the clr-transformed microbial abundance and factor (x axis). The lower red bars show the denominator taxa. The color tint is proportional to the reproducibility of the taxa in cross-validation analysis. (**B**) Relation between the balance composed of reproducible taxa and lymphocyte count (*n* = 177). Red dots indicate values that exceeded the normal range.

**Figure 3 microorganisms-11-01036-f003:**
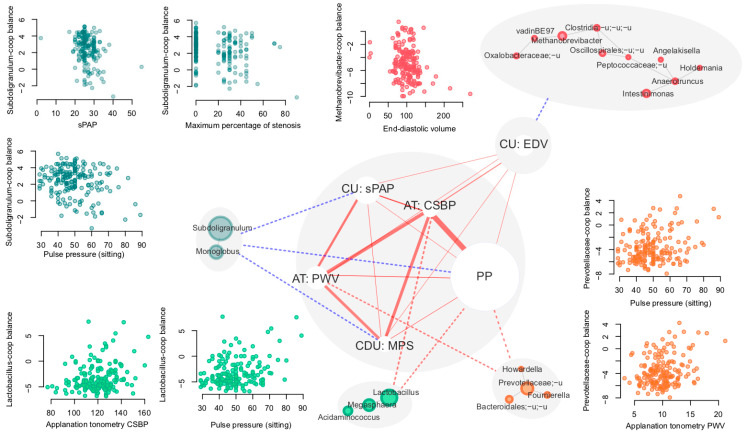
Associations between clinical cardiovascular markers and genera cooperatives. Significant correlations and associations are shown with solid and dashed lines, respectively. Red lines denote positive associations and blue lines denote negative associations. The width of the solid lines is proportional to the Spearman correlation coefficient between cardiovascular markers (ranging from 0.17 to 0.54). The size of nodes corresponding to markers is proportional to the minimal *p* value of the associations between the marker and cooperatives. The size of the nodes corresponding to genera is proportional to their mean abundance in the dataset.

**Figure 4 microorganisms-11-01036-f004:**
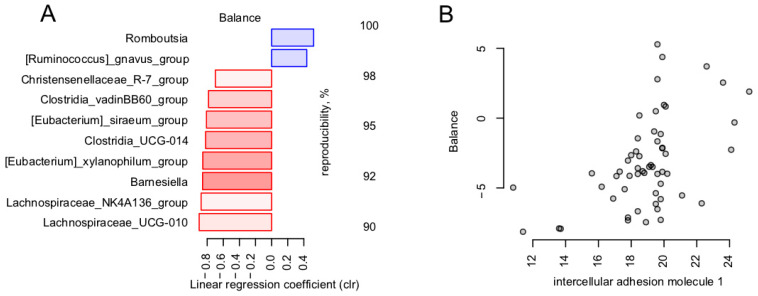
Link between endothelial dysfunction markers and gut microbiome in post-COVID-19. Reproducible taxa were revealed by nearest balance cross-validation associated with ICAM-1 (ng/mL). Constructed similarly to Figure 2: (**A**) Reproducible members of the nearest balances. (**B**) Relation between the balance composed of reproducible taxa and ICAM-1 (*n* = 56).

**Figure 5 microorganisms-11-01036-f005:**
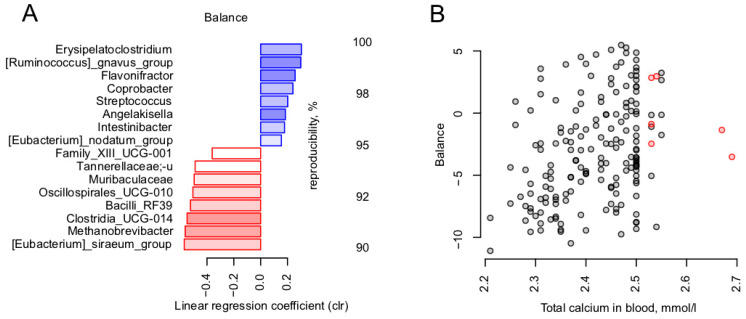
Link between blood metabolites and gut microbiome in post-COVID-19. Reproducible taxa were revealed by nearest balance cross-validation associated with total calcium blood level, mmol/L. Constructed similarly to Figure 2: (**A**) Reproducible members of the nearest balances. (**B**) Relation between the balance composed of reproducible taxa and total calcium in blood, mmol/L (*n* = 180).

**Table 1 microorganisms-11-01036-t001:** Group-wise statistics for the cardiovascular parameters.

Parameter	Units	Group A (Med, Q1; Q3)	Group N (Med, Q1; Q3)	Group S (Med, Q1; Q3)
Ultrasound dopplerography of the carotid arteries				
Intima media thickness right	cm	0.73; 0.64, 0.84 (*n* = 46)	0.73; 0.64, 0.88 (*n* = 48)	0.88; 0.74, 0.98 (*n* = 86)
Maximal stenosis		0; 0, 30 (*n* = 46)	0; 0, 25 (*n* = 48)	25; 0, 35 (*n* = 86)
ECHO-CG				
Myocardial mass index	g/m^2^	62.5; 55, 73.83 (*n* = 42)	64.5; 54.25, 74 (*n* = 46)	75; 62.25, 88.75 (*n* = 74)
Anteroposterior dimension of the right ventricle	cm	2.9; 2.7, 3.1 (*n* = 46)	2.8; 2.6, 3.1 (*n* = 45)	3.1; 2.8, 3.4 (*n* = 85)
Left atrial volume	mL	44.5; 38.25, 54.75 (*n* = 46)	48; 36.5, 57 (*n* = 47)	56; 45.5, 68.5 (*n* = 83)
Right atrial volume	mL	33.5; 28, 39.5 (*n* = 46)	32; 27, 40 (*n* = 47)	36; 32.25, 43.75 (*n* = 82)
Systolic pressure in the pulmonary artery	mmHg	25; 24, 27 (*n* = 46)	25; 24, 30 (*n* = 47)	27; 25, 30 (*n* = 85)
E/A ratio		1.27; 0.85, 1.6 (*n* = 46)	1.3; 0.97, 1.5 (*n* = 47)	0.8; 0.7, 1 (*n* = 81)
E/e’ ratio		6.5; 5, 7 (*n* = 46)	7; 6, 8 (*n* = 47)	8; 7, 10 (*n* = 80)
Left ventricular diastolic dysfunction		2; 1, 2 (*n* = 46)	2; 1, 2 (*n* = 47)	2; 2, 2 (*n* = 82)
End diastolic volume	mL	92; 82.75, 107.5 (*n* = 46)	88; 71.25, 109.5 (*n* = 46)	108; 92, 118 (*n* = 85)
Ejection fraction		60.5; 60, 64 (*n* = 46)	63; 60, 64.5 (*n* = 47)	60; 58, 62 (*n* = 85)
Applanation tonometry				
Pulse Wave Velocity	m/s	9.7; 7.9, 10.67 (*n* = 46)	8.6; 6.8, 10.3 (*n* = 44)	10.5; 9.1, 12.1 (*n* = 85)
Central systolic blood pressure	mmHg	117; 108.25, 124.75 (*n* = 46)	113.5; 106.25, 123.5 (*n* = 46)	125; 115.25, 134 (*n* = 86)
Central diastolic blood pressure	mmHg	79.5; 71.25, 84.75 (*n* = 46)	76; 71, 81.75 (*n* = 46)	86; 79.25, 91 (*n* = 86)
Augmentation Index		25; 14, 29 (*n* = 46)	21; 13.25, 30.25 (*n* = 46)	24.5; 17.25, 30 (*n* = 86)
Sitting heart rate	bpm	71; 64, 76 (*n* = 46)	68; 65, 77 (*n* = 48)	72; 66, 78 (*n* = 86)
Sitting pulse blood pressure	mmHg	45.5; 41.25, 52.75 (*n* = 46)	45; 40, 50 (*n* = 48)	50; 44, 60 (*n* = 86)
Concentric remodeling	Yes|No	10|36 (*n* = 46)	19|28 (*n* = 47)	32|53 (*n* = 85)
Degree of mitral regurgitation	1st|2nd	37|8 (*n* = 45)	41|6 (*n* = 47)	50|34 (*n* = 84)

**Table 2 microorganisms-11-01036-t002:** Cardiovascular markers significantly linked to gut microbiome composition at the level of bacterial cooperatives. Samples are pooled across the groups.

Cooperative Used as a Balance Numerator	Factor	Linear Model Coefficient	*p*	FDR	Number of Subjects
Lactobacillus-coop	Pulse pressure in sitting position (PP)	0.8576	0	0.0034	180
Lactobacillus-coop	Central systolic blood pressure (CSBP)	0.6591	0.0009	0.0468	178
Prevotellaceae-coop	Pulse Wave Velocity (PWV)	0.6488	0.0003	0.0357	175
Prevotellaceae-coop	Pulse pressure in sitting position (PP)	0.5846	0.001	0.0468	180
Subdoligranulum-coop	Pulse pressure in sitting position (PP)	−0.4016	0.0018	0.0703	180
Subdoligranulum-coop	Maximum percentage of stenosis (MPS)	−0.427	0.0009	0.0468	180
Subdoligranulum-coop	Systolic pulmonary artery pressure (sPAP) (highly correlated with*:* tricuspid regurgitation gradient) *	−0.4658	0.0002	0.0351	178
Methanobrevibacter-coop	End-diastolic volume (EDV) (highly correlated with*:* end-diastolic size) *	−0.6481	0.0004	0.0357	177

* for each pairwise highly correlated clusters of parameters (|rho| > 0.8, see Section 2), one parameter per cluster was chosen for the analysis.

**Table 3 microorganisms-11-01036-t003:** Group-wise statistics for the endothelial dysfunction parameters.

Parameter	Group N (Med, Q1; Q3)	Group A (Med, Q1; Q3)	Group S (Med, Q1; Q3)
VEGF-R1	0.2; 0.14, 0.26 (*n* = 42)	0.18; 0.14, 0.25 (*n* = 47)	0.19; 0.13, 0.28 (*n* = 79)
ICAM-1	19.6; 17.8, 19.8 (*n* = 9)	18.2; 17.65, 19.75 (*n* = 14)	19.5; 18.5, 19.9 (*n* = 33)
VCAM-1	12.65; 8.54, 27.3 (*n* = 42)	11.4; 8.21, 31.8 (*n* = 47)	19.7; 10.15, 38.45 (*n* = 79)
E-selectin	7.32; 5.8, 11.38 (*n* = 42)	8.1; 5.08, 12.4 (*n* = 47)	8.38; 6.27, 12.6 (*n* = 79)
von Willebrand factor, %	103.5; 85.5, 122.5 (*n* = 46)	116; 93, 145 (*n* = 46)	143; 117, 177 (*n* = 85)

## Data Availability

The sequencing data are available under accession number PRJNA875118, https://dataview.ncbi.nlm.nih.gov/object/PRJNA875118?reviewer=4gmgp9s9g36lo1loljd1ctnbla, accessed on 1 March 2023.

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
