# Peer review of "Gut Microbiome in Post-COVID-19 Patients Is Linked to Immune and Cardiovascular Health Status but Not COVID-19 Severity"

_microorganisms, 2023, doi:10.3390/microorganisms11041036_

Round 1

Reviewer 1 Report

In this study, the authors investigated the microbiome 3 months after the end of the disease in patients who got asymptomatic COVID-19 (A), severe COVID-19 (S) or in patients who did not get infected (N) and compared microbiota composition with clinical parameters. Their results suggest comparable recovery of the gut microbiota 3 months post-COVID-19 independently of the severity of the infection (A versus S). But they observed multiple associations with clinical data, such as relative lymphocytes number, cardiovascular parameters and blood calcium levels.

The topic is important as the gut microbiota is influencing various diseases as described in the introduction. Additional data such as the ones proposed in this study, are needed to increase our understanding of the implication of gut microbiome in COVID-19.

Line 19. .. after the end of the disease in 178 post-COVID-19 patients.

It seems to be not correctly stated. While group N (n=46) is an interesting group to compare to (“resistant’’) they did not develop COVID-19, so there was no disease, and they cannot be called post-COVID-19 patients.

Similarly, line 230. Three months after infection, .. subjects who contacted with COVID-19 patients with no further infection.. Should be reformulated, because no differences in microbiome structure could be assessed in group A versus S 3 months after recovery but not in group N as there were not infected. Microbiota can be compared with group N then,

Please coherently state along the manuscript three months after infection (line 230) or 3 months after the end of the disease (line 19)

Line 20. .line 232. ... who contacted COVID-19 patients with no further infection.

Maybe reformulate. “who got in contact with COVID-19 patients but did not get infected (“resistant”)”

Line 42.  .. an important issue today is the study of the factors underlying the risk of severe disease and post-infection complications.

Based on the methods (3 months after recovery) of this study, no information would be presented regarding risk factors for severity, hence I would reformulate “factors underlying the risk of severe disease and post-infection complications.

Line 48. If reference 5 the correct one?

Line 68. Add reference

Line 71. Reference is missing.

Line 75. Authors Zuo et al. are missing in reference 21.

Line 77. Add reference after the sentence.

Line 80. Chen et al, 2022, Gut, “Six-month follow-up of gut microbiota richness” found that microbiota richness was not restored to normal levels after 6-month recovery and Tian et al, 2021, “Gut microbiota may not be fully restored after 3 months” in a cohort of 7 patients.

To be discussed here.

In addition, it might be interesting to mention the study of Liu et al, 2022 “Gut microbiota dynamics in a prospective cohort of patients with post-acute COVID-19 syndrome”. There the authors showed that patients without long COVID showed recovered gut microbiome profile 6 months after comparable to non-COVID-19 controls but gut microbiota of long COVID patients was altered.

Do the authors have information regarding the long COVID symptoms of their cohort ?

This could be potentially added in the discussion as well.

Line 348. Add to ref 34 (30 days after disease resolution), the reference Zhang et al, 2022 “prolonged impairment of SCFA in gut microbiome in COVID-19 patients”  (analysis done also 1 month after recovery)

References 49, 54, authors are missing.

Author Response

We thank the reviewer for the careful reading of the manuscript and constructive remarks. We have taken the comments on board to improve and clarify the manuscript. Please find below a point-by-point response to all comments.

  • Line 19. .. - Corrected
  • Similarly, line 230… - Corrected
  • Please coherently state along… - Corrected
  • Line 20. .line 232… - Corrected
  • Line 42.  … - Corrected
  • Line 48… Yes, it’s correct.
  • Line 68… We’ve checked it, reference [24] is cited
  • Line 71… - Corrected
  • Line 75… - Reference is corrected
  • Line 77… - Added
  • Line 80… - We’ve added some information and cited mentioned study
  • In addition… We’ve mentioned these studies as well
  • Do the authors have… Thank you for your comment. We have some data on long-COVID. Due to the fact that this issue requires a detailed description of diagnostics and statistical analysis, we decided not to overload this article with additional results, but in the future we will describe this information in another article
  • Line 348… - Added
  • References 49, 54… - Fixed

Reviewer 2 Report

The paper is interesting and is structured quite well. The subject matter is very topical.

Author Response

Dear reviewer,

Thank you for your work and positive comments on the manuscript. 

Reviewer 3 Report

Comments and Suggestions for Authors

The authors Tkacheva et al. in the manuscript  entitled “Gut microbiome in post-COVID-19 patients is linked to immune and cardiovascular health status but not COVID-19 severity”, highlight through statistical analysis the correlation between some biochemical, immune and  cardiovascular parameters and the microbiome of COVID-19 patients three months after the disease event. They note in conclusion that the microbiome has the potential to heal and that among the three groups analyzed, there were no differences in diversity of composition.

It should be pointed out that the study groups reflect a homogeneous geographic area and population. That being said, their manuscript is very important, and having resolved the major and minor revisions is worthy of publication.

Important: The manuscript needs a thorough high resolution of graphs. 

 Major revisions:

1.      Introduction: Please Extend more to the introductory part explaining both airborne but also oro-fecal routes of virus transmission

2.      Lines 56-58: Also related to changes or alteration of the microbiome are other diseases such as Parkinson's, Alzheimer, frontotemporal dementia, and breast cancer, colon cancer. Please also extend the concept to these correlations by citing references.

3.      Line 63: what was written about antibiotics is true on the one hand because they can affect the microbiome by depriving it of important bacteria to regulate surface immunity, but the opposite is also true that if taken at the exact time, they can have the opposite effect. I advise the authors to also cite opposing arguments, as in the present paper:

a.     Woods Acevedo, M. A., & Pfeiffer, J. K. (2020). Microbiota-independent antiviral effects of antibiotics on poliovirus and coxsackievirus. Virology, 546, 20–24. https://doi.org/10.1016/j.virol.2020.04.001

4.      Lines 68-74: This paragraph is important. The authors have described a very good job. I also recommend reading and arguing other studies highlighting the connexion between bacteria and SARS-CoV-2:

a.     Brogna, C.; Costanzo, V.; Brogna, B.; Bisaccia, D.R.; Brogna, G.; Giuliano, M.; Montano, L.; Viduto, V.; Cristoni, S.; Fabrowski, M.; Piscopo, M. Analysis of Bacteriophage Behavior of a Human RNA Virus, SARS-CoV-2, through the Integrated Approach of Immunofluorescence Microscopy, Proteomics and D-Amino Acid Quantification. Int. J. Mol. Sci. 2023, 24, 3929. https://doi.org/10.3390/ijms24043929

5.      Lines 77 -80: please also comment on other authors who find the opposite, for example, a very important difference in the abundance of Faecalibacterium prausnitzii or other healthy bacteria:

a.     Zuo, Tao et al. “Alterations in Gut Microbiota of Patients With COVID-19 During Time of Hospitalization.” Gastroenterology vol. 159,3 (2020): 944-955.e8. doi:10.1053/j.gastro.2020.05.048

6.      Line 106-126: Are molecular tests for COVID sufferers, not real-time RT-PCR? What do the authors mean by PCR? SARS CoV 2 is an RNA virus.

7.      Line 178: trimmed according to the 178 primers used (515F/806R§). Please add references

8.      Line 182: Shannon and chao1, please add references

9.      Line 185: Aitchison distance metric, please add references

10. Line 224: Gut microbiome composition is not associated neither with resistance to SARS-CoV-2 nor disease severity….. it should be considered that the study did not analyze whether the bacteria had integrated or modified sequences in the whole genome or in plasmids of viral origin, as there are studies showing that SARS-CoV-2 has bacteriophagic behavior,  and the authors at  line 247 and 248 clarifies that the analyses were done 3 months after the event, it is appropriate to correct the title in a more scientifically correct manner: “Gut microbiome composition is not associated neither with the severity of  disease after 3 mouths

11.   Line 241, Figure 1: please add a legend to N, A, S

12.   For figures 2-5, please replace figures with another at a higher resolution.

Minor revision

1.     Line 38: “and in a wave-like fashion”.. Please add some references

2.     Lines 38-40: Enter the date when the 680 million figure was taken and the WHO page reference

3.     Line 146: please indicate what type of antibodies  were observed in the patients, whether toward the spike protein or toward the nucleocapsid protein and indicate the brand of reagent used and the antibodies

4.     Line 132: correct by specifying BLOOD oxygen saturation

5.     Lines 134 and 152: CT, ABP, HR, RR, BMI, CBC, WBC, and everywhere, all abbreviated words should state the first time they appear in the text first the full name, and then the abbreviation can be used. For example, BMI, write the first-time body mass index ( BMI)

6.     Line 165: “sample collection and processing”: Describe the process briefly, in a summarized manner, to ar that the reader can read it in this manuscript without going looking for it in the one already published

7.     Line 171: the link provided cannot be found on the web; please correct it.

8.     Line 190: why did the authors decide to use these criteria> 20 reads? Please give an explanation. don't the authors think that even small details can be useful information for research? Or did they perform like other studies? If so add a reference.

9.     Lines 195-200: Please list the parameters analyzed in the supplementary materials

10.   Line 396: Please add reference

Author Response

We thank you for the careful review of our manuscript and thoughtful feedback. The comments have been very thorough and useful in improving the manuscript. Our point-by-point response to the comments are given below:

  • Important: The manuscript needs a thorough high resolution of graphs. - All pics were reuploaded to the system.
  • 1.      Introduction… Corrected
  • 2.      Lines 56-58… Added
  • 3.      Line 63… Corrected
  • 4.      Lines 68-74… Thanks for the interesting source. The article is really interesting, but we decided not to add materials from it, so as not to overload our manuscript.
  • 5.      Lines 77 -80… Commented
  • 6.      Line 106-126… Thank you for your comment, it was RT-PCR, corrected.
  • 7.      Line 178… Reference was added
  • 8.      Line 182… References were added
  • 9.      Line 185… Reference was added
  • 10.    Line 224… Corrected
  • 11.    Line 241… Added
  • 12.    For figures 2-5…We reuploaded figures
  • 1.      Line 38… Corrected
  • 2.      Lines 38-40… Corrected, source is added
  • 3.     Line 146…Specified
  • 4.     Line 132… Specified
  • 5.     Lines 134 and 152… Full names were added 
  • 6.     Line 165… Described
  • 7.     Line 171… Link was updated 
  • 8.     Line 190… It’s one of the commonly used thresholds to filter out noise. We’ve cited it in the manuscript.
  • 9.     Lines 195-200… Added to supplementary 

10.   Line 396… We’ve corrected the sentence to make it more clear.

Reviewer 4 Report

This article is well written. It suggests that comparable recovery of the  gut community structure in post-COVID-19 period independently of the severity or infection status. Multiple identified associations of  microbiome provide hypotheses about participation  in regulating immunity and homeostasis of cardiovascular and other  body systems in health, as well as their disruption in SARS-CoV-2 infections and other diseases. 

It would be appealing for the readers if changes were made 

1. could include  pathway linking alteration in gut microbiota  and cardiometabolic disease 

2. As hypertension, Diabetes, Dyslipidemia and smoking are the major risk factors, could include that in the demographics among the selected population ( asymptomatic/ Non-infected and severe patients) 

3. Gut microbiota composition, Gut microbiota metabolites  and possible concepts linked to cardiac disease 

Author Response

Dear reviewer,

Thank you for the comments on our manuscript. We sincerely appreciate your valuable suggestions, which helped us in improving the quality of the manuscript.

We added a table with risk factors in groups to supplementary materials and added some corrections to the document.

Information about the microbiome metabolic potential obtained from amplicon sequencing data with PICRUSt or similar software usually can't provide much insights in addition to taxonomy analysis. The reason is high variability of bacteria genomes content and therefore persize metabolic potential prediction. Therefore significant metabolites in such data usually reflect taxonomic associations at high ranks (phylum, class, order).

Round 2

Reviewer 3 Report

The authors have resolved all my suggestions.

Reviewer 4 Report

Thank you for editing your article. I appreciate adding the suggestions.